# Acute Aortic Stent Graft Thrombosis in Patient with Recent COVID-19 Infection

**DOI:** 10.3390/reports7010004

**Published:** 2024-01-12

**Authors:** Antonio Marzano, Jihad Jabbour, Vincenzo Brizzi, Enrico Sbarigia, Simone Cuozzo

**Affiliations:** 1Vascular Surgery Division, Department of Surgery “Paride Stefanini” Policlinico Umberto I, “La Sapienza” University of Rome, Viale del Policlinico 155, 00161 Rome, Italy; drjihadjabbour@gmail.com (J.J.); enrico.sbarigia@uniroma1.it (E.S.); simone.cuozzo89@gmail.com (S.C.); 2Vascular Surgery Department, Centre Hospitalier Universitaire de Bordeaux, 33404 Bordeaux, France

**Keywords:** EVAR, COVID-19, stent graft thrombosis, explantation

## Abstract

Although COVID-19 primarily affects the respiratory system, it can have various effects on other organs, including the cardiovascular system. COVID-19 can lead to a prothrombotic status, promoting blood clotting, which can potentially affect native vessels and implanted devices. The exact mechanisms through which it leads to increased clotting are not yet fully understood but may involve inflammation, endothelial dysfunction, and a hyperactive immune response. In the present report, we describe a case of acute aortic stent graft thrombosis four days after the resolution of SARS-CoV-2 infection. The patient required emergent explantation of the stent graft after the failure of endovascular bailout procedures.

## 1. Introduction

As described in the literature [1,2], COVID-19 infection burdens people with a high rate of thrombotic complications. The status of increased blood hypercoagulability, probably due to systemic hyper-inflammation and endothelial dysfunction, is associated with both venous and arterial thrombosis [3,4]. Arterial thrombosis accounts for about 4% of thromboembolic complications during COVID-19. Native aortic acute thrombosis related to severe acute respiratory syndrome coronavirus 2 (SARS-CoV2) infection has been largely described [5,6], but, to our knowledge, no case of acute aortic stent graft occlusion has been previously described in the literature. Moreover, there are no clear international guidelines regarding peri-operative management of patients with SARS-CoV-2, even though it has been four years since it appeared.

We report a case of a patient recently treated for abdominal aorto-iliac aneurysm by EVAR with Bolton Treo stent graft implantation (Bolton Medical Inc., Sunrise, FL, USA), four days after the resolution of a paucisymptomatic SARS-CoV-2 infection. At the time of emergency admission, physical examination showed acute limb ischemia that was confirmed by computed tomography angiography (CTA), which demonstrated complete thrombosis of the aortic stent graft.

## 2. Detailed Case Description 

Elective EVAR by Bolton Treo stent graft implantation for aorto-iliac aneurysm through bilateral percutaneous common femoral artery (CFA) access was planned in a 73-year-old male patient. The patient’s medical history reported arterial hypertension, hypercholesterolemia, chronic obstructive pulmonary disease, previous open abdominal surgery, and a right popliteal artery aneurysm with a maximum transversal diameter (DT max) of 23 mm. Lower limb pulses were present at the physical examination.

Although the AAA’s DT max was <55 mm, the patient presented a bilateral common iliac artery aneurysm with the DT max up to 35 mm (right 32 mm, left 38 mm), associated with severe stenosis of the right internal iliac artery (IIA) and chronic total occlusion of the left IIA (Figure 1a). As is known, aortic outflow occlusion is related to a major risk of aneurysm rupture and early elective repair is recommended to minimize risks in these cases [7].

The sizing of the endoprosthesis was selected following the manufacturer’s instructions for use (IFU). The proximal aortic neck was 24 mm in diameter and 15 mm in length; therefore, a 28–80 mm Bolton Treo was selected.

Based on the anatomical findings of preoperative CTA, the distal sealing zone was obtained at the level of the external iliac arteries (EIAs). The iliac artery tortuosity index [8] (X) and EIA diameters are listed in Table 1.

Sac embolization with coils was deemed necessary to prevent type II endoleak due to the large diameter (4.8 mm) of the inferior mesenteric artery (IMA) and numerous and large lumbar arteries (Figure 1b) [9]. Sac embolization was performed via left femoral access with coaxial catheter insertion into the aneurysmal sac with subsequent release of two semi-controlled Interlock coils (Boston Scientific, Marlborough, MA, USA) after EVAR completion.

Systemic heparinization with a bolus of 5000 IU unfractionated heparin was performed, as our protocol, immediately after percutaneous femoral accesses. The procedure lasted 100 min, without technical difficulties. Completion angiography showed the proper deployment of the endograft, the absence of endoleaks, and excellent outflow at the level of EIAs. No groin complications occurred.

The EVAR procedure was performed four days after the resolution of paucisymptomatic SARS-CoV-2 infection, confirmed through reverse transcriptase-polymerase chain analysis, performed due the presence of a COVID-19 infection cluster in our Surgical Department. The patient’s infection lasted for up to ten days and was characterized by fever, never exceeding 38 °C, for only four days. No COVID-19-specific therapies were deemed necessary.

During the hospitalization, the patient experienced non-specific lower back pain. As there were no other definite causes for it, it seemed more prudent to treat the AAA rather than discharge the patient.

Preoperative oxygen saturation (SaO_2_) was 98% in air and arterial blood gas (ABG) in air showed pH 7.44, PaCO_2_ 29 mmHg, PaO_2_ 66 mmHg, and P/F ratio 314. Blood tests showed normal platelet count (154 × 10^9^/L), fibrinogen 0.22 g/dL, D-dimer 500 ng/mL, and hs-CRP 4.7 mg/L. The post-operative course was regular but characterized by post-implantation syndrome (PSI). Although no fever ≥ 38 °C was observed, the leukocyte count was >12,000/mL and hs-CPR was >10 mg/L. After two days, blood tests showed a normal leukocyte count and a significative trend towards hs-CRP. As per our protocol, a duplex ultrasound examination (DUS) was performed before the discharge. No pseudoaneurysms or arterio-venous fistulas were detected at the level of femoral access; a normal flow pattern upstream, within, and downstream of the stent graft was detected.

At the physical examination, all of the arterial pulses at the lower extremities were present.

The patient was discharged on the third post-operative day under antiplatelet therapy.

After twenty days, the patient was admitted to our Emergency Department reporting with pallor, pain, and hypothermia of the lower limbs. A sensory and motor deficit of the foot and legs was present (Rutherford class IIB). Blood tests showed fibrinogen > 0.55 g/dL (range, 0.2–0.4), D-dimer > 4509 ng/mL (range, 0–550), creatine phosphokinase 16,623 IU/L, myoglobin > 30,000 ng/mL, lactate dehydrogenase 256 IU/L, IL-6 > 200 pg/mL (range, 2–6), and hs-CRP 24.7 mg/L. These values are collected in Table 2.

A bedside DUS examination showed the patency of the suprarenal aorta and complete thrombosis of the aortic stent graft without detectable blood flow at the femoral level.

An urgent CTA confirmed complete endograft thrombosis with common femoral arteries’ reconstitution through collateral circulations (Figure 2) and thrombosis of the right popliteal artery aneurysm.

After consulting a team of interventional radiologists, an endovascular treatment with fibrinolytic agent infusion was preferred. Catheter-directed thrombolysis with urokinase infusion (60,000 IU/h) was started through bilateral percutaneous CFA access. No fabric infolding, stent fracture, or stenosis of the native EIAs were noted (Figure 3).

Although fibrinolytic infusion almost dissolved the entire clot, residual floating thrombosis of the main body of the stent graft was observed in the control CTA (Figure 4).

Mechanical thrombectomy with AngioJetTM (Possis, Minneapolis, MN, USA) was attempted but it failed because of the tenacious adhesion of the residual clot to the fabric (Figure 5).

Therefore, proximal aortic cuff deployment to exclude the floating thrombosis was planned. Unfortunately, the cuff was deployed across the free-flow’s stents of the bifurcated stent graft. Thus, any additional bailout endovascular procedures were deemed unfeasible and the patient underwent an urgent explantation and an aorto-bifemoral prosthetic bypass. After the midline incision was performed, a non-pulsatile mass was observed. The mobilization of the left renal vein allowed the dissection of the suprarenal aorta. A suprarenal aortic cross clamp was placed above the renal arteries. The aneurysmal sac was opened and the endograft was exposed. The bifurcated stent graft was cross-sectioned just above its bifurcation and iliac limbs were clamped. The proximal aortic cuff was distally pulled and entirely removed without difficulty, as most of the main body of the previous stent-graft. It was not deemed necessary to completely remove the free-flow of the bifurcated stent graft as there was no infection. Subsequently, the proximal anastomosis with 18 mm × 9 mm Dacron prosthesis was performed. The duration of the suprarenal clamp was about 20 min.

Mechanical thrombectomy of the thrombosed right popliteal artery aneurysm with Penumbra/Indigo Systems (Penumbra Inc., Alameda, CA, USA) was performed after the restoration of the aortic flow. A Gore Viabahn 6–50 mm (WL Gore & Associates, Flagstaff, AZ, USA) was deployed to exclude the aneurysm and to prevent distal embolization.

After the procedure, fasciotomies were performed.

The patient was transferred to the Intensive Care Unit (ICU) for three days. No complications occurred during the post-operative period and the patient was discharged from the hospital twelve days later, under dual antiplatelet therapy.

Four months after open conversion, a follow-up CTA showed the patency of the prosthetic aortic bypass and complete exclusion of the right popliteal aneurysm.

Written informed consent was obtained from the patient to publish this paper.

## 3. Discussion

COVID-19 can manifest with a wide variety of symptoms, including very serious ones, due to the possible involvement of various systems such as cardiovascular, gastrointestinal, nervous, and musculoskeletal.

Coagulopathy and hypercoagulation are widely known severe complications of COVID-19 infection which can have unfavorable outcomes and lead to a high mortality rate. Approximately one third of patients with COVID-19 infection may experience venous or arterial thrombosis [10]; this state has been termed COVID-19-associated coagulopathy (CAC) [11].

The estimated incidence of arterial thrombosis during SARS-CoV-2 infection is about 4%, but coagulopathy may persist even after the acute phase of infection and even after negativity [12,13].

Arterial thrombosis more often involves lower limb arteries than large vessels, whereas prosthetic graft [14] and stent graft thrombosis is rare. Aortic stent graft thrombosis is favored by concurrent conditions such as stenosis or severe occlusive disease of outflow vessels [15,16].

The pathogenesis of hypercoagulability in COVID-19 is ill defined. All three components of Virchow’s triad appear to be involved, including endothelial injury, stasis, and hypercoagulable state.

Endothelial injury is evident from the direct invasion of endothelial cells by SARS-CoV-2; endothelial cells have a high number of angiotensin-converting enzyme 2 (ACE-2) receptors. SARS-CoV-2 enters the cell through the ACE-2 receptor [17]. Increased angiogenesis was also reported in these patients [18]. Increased cytokines are released, such as interleukin (IL-6), and various acute-phase reactants in COVID-19 can lead to endothelial injury [19]. Moreover, the use of intravascular catheters can cause direct endothelial cell injury too. Stasis is due to immobilization in all hospitalized patients, especially those who are critically ill. A hypercoagulable state is seen due to several coagulation abnormalities from elevated circulating prothrombotic factors such as elevated von Willebrand factor (vWF), factor VIII, D-dimer, fibrinogen, neutrophil extracellular traps, prothrombotic microparticles, and anionic phospholipids [20].

We analyzed several variables to determine the etiology of this complication. No native EIA stenosis was detected at preoperative CTA, at completion angiography, or at post-operative CFA Doppler waveform analysis. Before admission to the Emergency Department, the patient did not experience intermittent or buttock claudication. The most recent outpatient examination demonstrated the patency of the stent graft without significant distal disease.

Moreover, no evidence of compromise to the integrity of the device was observed. We also excluded congenital and acquired thrombophilia, as well as infection, as the potential causes of thrombosis because laboratory studies for these conditions were negative.

Consequently, the possible explanation of acute stent graft thrombosis was hypercoagulability and systemic inflammation related to recent COVID-19 infection, as supported by laboratory tests [21]. In addition, thrombosis of popliteal aneurysms is reported in the literature because of SARS-CoV-2 infection [20].

Complete blood counts, inflammation, and coagulation tests were evaluated. A significant increase in IL-6, hs-CRP, fibrinogen, and D-dimer serum level was observed. Moreover, clot waveform analysis (CWA) showed a significantly higher aPTT, close to the upper limit of the reference range [22], characterized by increases in the density, velocity, and acceleration of clot formation. This higher density of clot can explain the tenacious adhesion to the fabric of the stent graft and the difficulties in completely removing the clot via fibrinolytic infusion and mechanical thrombectomy. It is well known that inflammatory systemic conditions, confirmed by the high serum level of inflammatory biomarkers, lead to interactions between neutrophils and platelets, such as platelet and complement activation. Although the platelet count remains normal, these inflammatory factors are collectively involved in immune-thrombosis, promoting thrombin-mediated fibrin generation and local blood clot formation.

In our case, another additional risk factor for thrombosis, compared with other COVID-19 patients, was the presence of the aortic stent graft, which may have been the ideal substrate to platelet aggregation. Some papers in the current available literature have investigated coronary stent thrombosis in patients with SARS-CoV-2 infection [23,24,25]. The mechanism involved in aortic graft thrombosis is most likely the same as that which occurs in coronary stents, with a much lower degree of incidence that could be explained by the larger diameter of an aortic graft compared to a coronary stent. Thus, the association between the thrombogenic mechanisms activated by COVID-19, which remain even months after the negativization, and the presence of intravascular devices seem to form the perfect combination capable of causing potentially catastrophic thrombosis.

Certainly, careful antiplatelet and anticoagulant therapy can balance, at least partially, high thrombotic risk. However, in a patient who has recently undergone surgery, attention must also be paid to the risk of bleeding.

Further experience and larger studies are needed to better define both etiology and treatment.

## 4. Conclusions

Although rare, acute aortic stent graft thrombosis is a possible and serious complication of COVID-19 infection.

No specific guidelines on medical treatment and surgical management are currently available regarding this condition, and the literature is mostly related to single-case reports, even though it has been four years since the appearance of SARS-CoV-2. In these cases, it would be advisable to delay aneurysm treatment and/or prescribe more aggressive antithrombotic therapy during and immediately after SARS-CoV-2 infection.

Despite the presence of less aggressive SARS-CoV-2 forms and increasingly updated vaccines, the need to continue to investigate aspects of COVID-19 that are still unclear remains fundamental given the high spread of the virus and the possibility of new pandemics caused by pathogens with characteristics similar to SARS-CoV-2 that remain.

## Figures and Tables

**Figure 1 reports-07-00004-f001:**
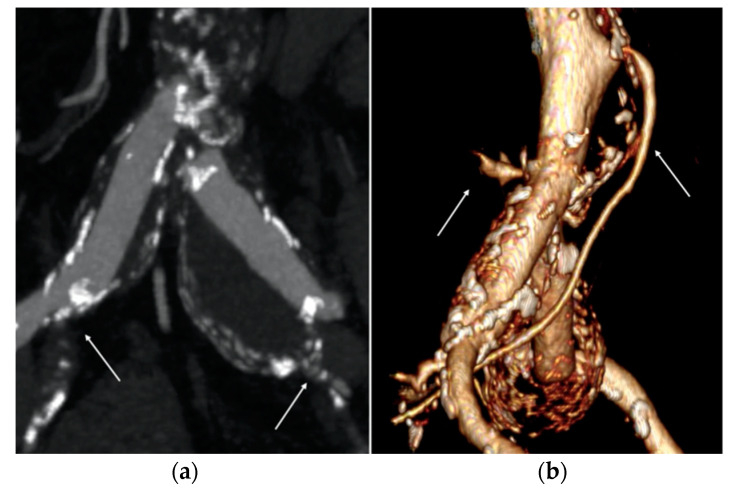
Pre-operative computed tomography angiography (CTA) images. (**a**) Right and left common iliac artery (CIA) aneurysms associated with severe stenosis of the right internal iliac artery and chronic total occlusion of the left one (white arrows). (**b**) Large diameter of the inferior mesenteric artery (IMA) and lumbar arteries (white arrows).

**Figure 2 reports-07-00004-f002:**
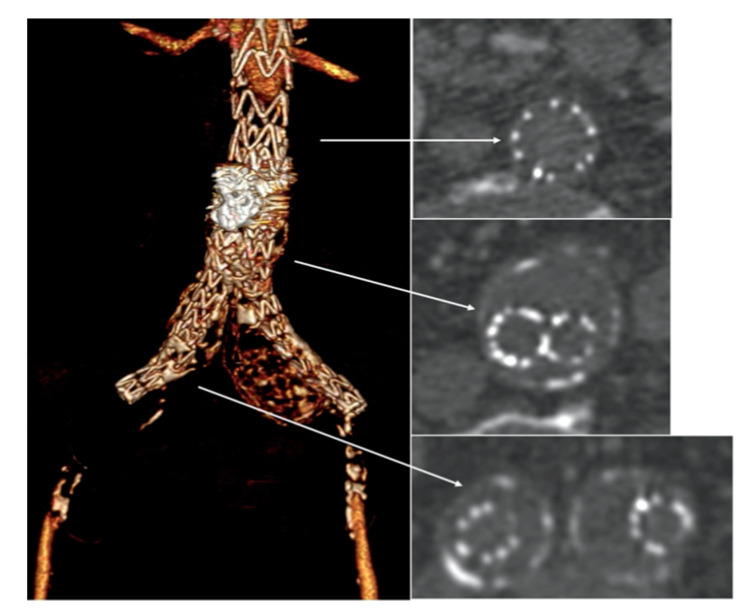
Emergency computed tomography angiography (CTA) showed complete endograft thrombosis (white arrows) with common femoral arteries’ reconstitution through collateral circulations.

**Figure 3 reports-07-00004-f003:**
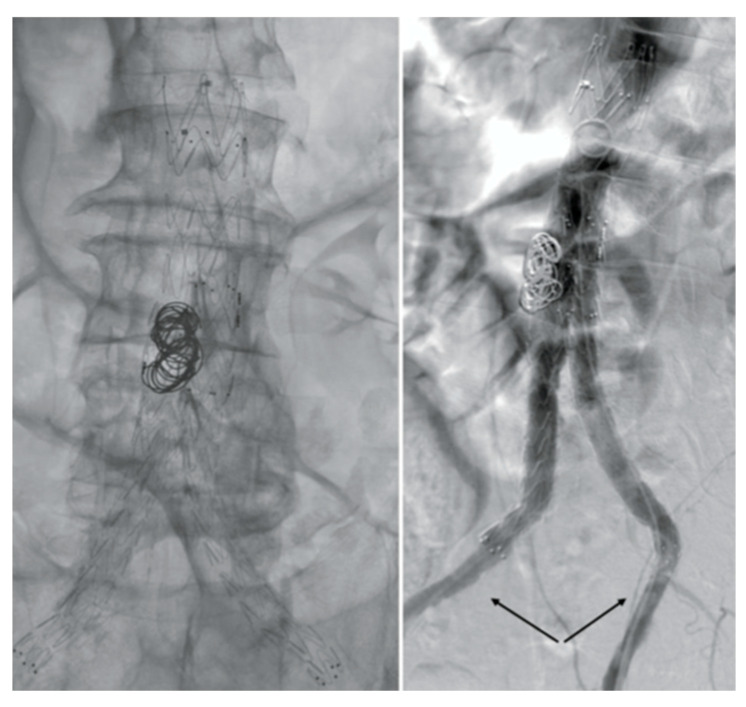
Angiography images. No fabric infolding, stent fractures, or stenosis of the native EIAs (black arrows) were noted.

**Figure 4 reports-07-00004-f004:**
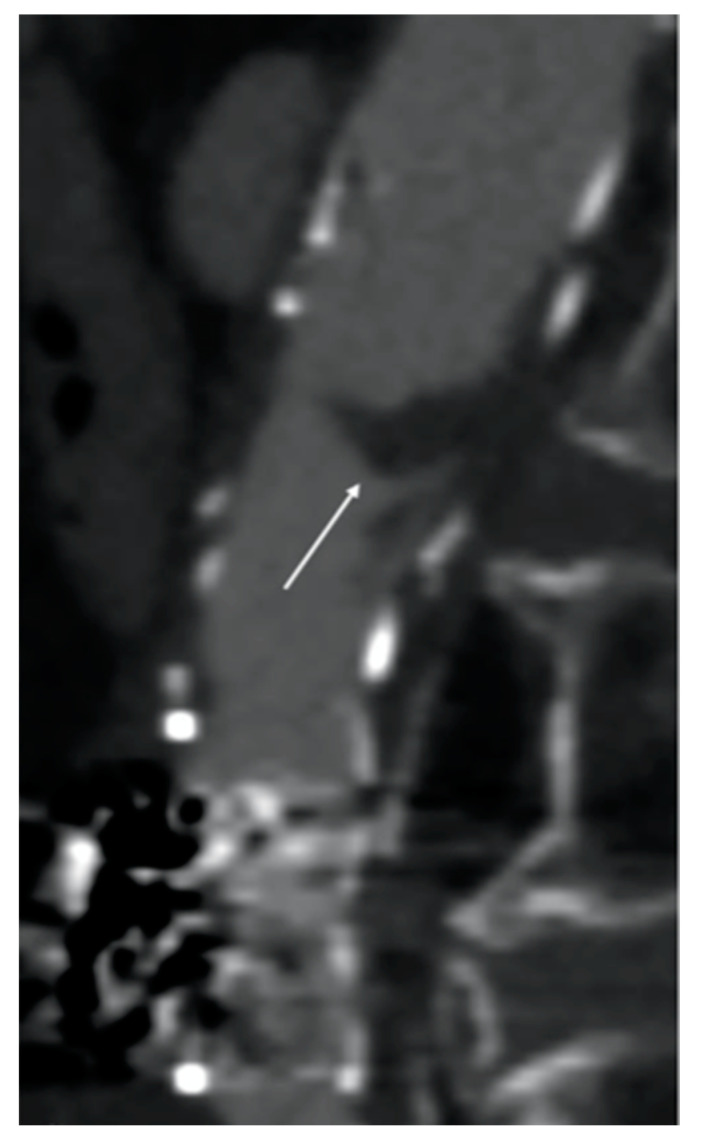
Computed tomography angiography (CTA) image after fibrinolytic agent infusion. Residual floating thrombosis (white arrow) of the main body of the stent graft was observed.

**Figure 5 reports-07-00004-f005:**
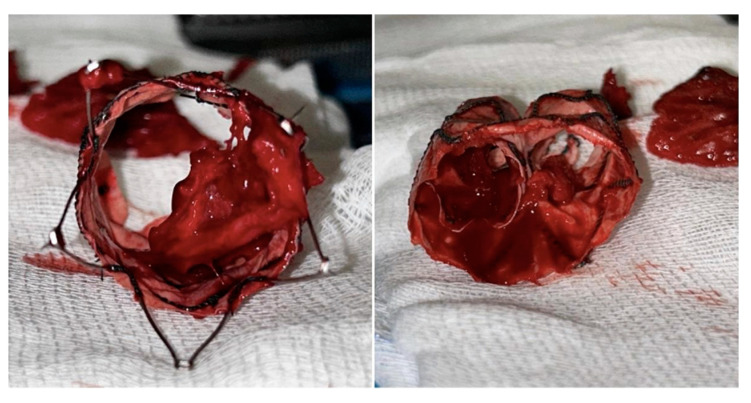
Intraoperative pictures of the tenacious adhesion of the residual clot to the fabric.

**Table 1 reports-07-00004-t001:** Iliac artery tortuosity index (X) and EIA diameters.

	Right	Left
Iliac artery tortuosity index (X)	1.14	1.17
External iliac artery diameter (mm)	8.8	7.8

**Table 2 reports-07-00004-t002:** Laboratory data upon admission to the Emergency Department.

Item	Value
Fibrinogen	>0.55 g/dL
D-dimer	>4 509 ng/mL
Creatine phosphokinase	16,623 IU/L
Myoglobin	>30,000 ng/mL
Lactate dehydrogenase	256 IU/L
Interleukin 6	>200 pg/mL
C-reactive protein	24.7 mg/L

## Data Availability

Data are contained within the article.

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
