# Peer review of "Acute Aortic Stent Graft Thrombosis in Patient with Recent COVID-19 Infection"

_reports, 2024, doi:10.3390/reports7010004_

Round 1

Reviewer 1 Report

Comments and Suggestions for Authors

Dear Authors,

I would like to thank you for the submission of a such interesting case. However you need to clarify and correct some issues in the manuscript.

Introduction, line 34: clinical suspicion was confirmed not only by CTA but also from his clinical picture of acute ischemia of both legs based on yuor manuscript. Please correct it.

Case Presentation, 40: You dont' refer the maximum diameter of AAA and of iliac aneurysm which is of paramount importance. An aneurysm of abdominal aorta less than 6 cm as you know has a very low probability of risk rupture and non specific back pain is not an absolute indication for operation in a recent positive Covid 19 patient. Why didn't you add a prophylactic LMWH together with aspirin for a short period after EVAR, especially in Covid 19 patient where thrombotic risk is higher than control group.

In such cases you must always refer that systemic heparinization was given (dose?) before performing an EVAR. This is missing in the text.Please check it

I don't agree with your strategy to land in both EIAs. You should try to recanalize the right IIA to improve perfusion on the pelvis and to extent the right limb to the CIA. Additionally you embolize too early the IMA which is an important collateral with IIAs especially in such a case with poor pelvic perfusion, with high risk of buttock claudication. You could wait at least 1 month to see if there is an aneurysm sac expansion. As you know type II endoleak could be resolved during follow up.

You don't descibe in details in the text the clinical picture of the patient after thrombosis of EVAR. What was the class of ischemia (Rutherford classification?

Please provide additional information regarding open conversion of the patient (suprarenal clamping ?, any trick to avoid damage of the renal arteries from the suprarenal fixation of the endograft)

Did the patient experience ischemic-reperfusion injury after restoration of blood flow to the extremities? Did you perform fasciotomies? Was the patient transferred to the ICU after the operation and for how long?

What was the platelet treatment of the patient after discharge?Viabahn endoprosthesis requires double antiplatelet treatment for at least 3 months. please refer

Discussion is well written describing the correlation of Covid 19 with thrombosis.

212 line. It is difficult to perform larger studies in patients with AAA treated with EVAR and recently positive Covid 19 patients. Please erase it

Author Response

Dear Reviewer,

thank you very much for your corrections and advice. The new manuscript has been implemented with all your suggestions.

Introduction, line 34: clinical suspicion was confirmed not only by CTA but also from his clinical picture of acute ischemia of both legs based on yuor manuscript. Please correct it.

Done

Case Presentation, 40: You dont' refer the maximum diameter of AAA and of iliac aneurysm which is of paramount importance. An aneurysm of abdominal aorta less than 6 cm as you know has a very low probability of risk rupture and non-specific back pain is not an absolute indication for operation in a recent positive Covid 19 patient. Why didn't you add a prophylactic LMWH together with aspirin for a short period after EVAR, especially in Covid 19 patient where thrombotic risk is higher than control group.

The maximum transversal diameter (DT max) of the AAA was smaller than 60 mm. The DT max of the common iliac arteries’ aneurysms was up to 35 mm (32 mm on the right, 37 mm on the left). The patient presented the severe stenosis of the right internal iliac artery and the chronic total occlusion of the left IIA. As known, aortic outflow occlusion is related to the risk of aneurysms rupture and an early elective repair is recommended in patients to minimize the risk of rupture. Moreover, isolated iliac arteries’ aneurysms treatment was deemed unfeasible due to the lack of proximal neck.

At time of case presentation, no guidelines or literature experiences were known about the useful use of prophylactic LMWH dose in this COVID 19 patients.

In such cases you must always refer that systemic heparinization was given (dose?) before performing an EVAR. This is missing in the text. Please check it

5000 UI, thank you.

I don't agree with your strategy to land in both EIAs. You should try to recanalize the right IIA to improve perfusion on the pelvis and to extent the right limb to the CIA. Additionally you embolize too early the IMA which is an important collateral with IIAs especially in such a case with poor pelvic perfusion, with high risk of buttock claudication. You could wait at least 1 month to see if there is an aneurysm sac expansion. As you know type II endoleak could be resolved during follow up.

The patient presented a severe stenosis of the right IIA. The mid- to long-term patency rate of stents in stenotic IIAs is not satisfying. Moreover, the length of the right CIA was <5 cm. An iliac branch device was not considered because these anatomical limitations. Moreover, due to SMA-IMA collateral circulation, sac embolization with coils did not induce an instant sac thrombosis, so we were confident to perform sac embolization at the same operative time, with reasonable risks of pelvic ischemia or buttock claudication. As described by Emboevar study, and as reported in other study (Aoki A, Maruta K, Omoto T, Masuda T. Midterm Outcomes of Endovascular Abdominal Aortic Aneurysm Repair with Prevention of type 2 Endoleak by Intraoperative Aortic Side Branch Coil Embolization. Ann Vasc Surg. 2022 Jan;78:180-189. doi: 10.1016/j.avsg.2021.06.037. Epub 2021 Sep 17. PMID: 34537351), intraoperative sac embolization could be performed in case of large or numerous LA o IMA (>2mm).

You don't descibe in details in the text the clinical picture of the patient after thrombosis of EVAR. What was the class of ischemia (Rutherford classification?)

Rutherford class IIb, Done. Thank you for suggestion.

Please provide additional information regarding open conversion of the patient (suprarenal clamping ?, any trick to avoid damage of the renal arteries from the suprarenal fixation of the endograft)

Done. Thank you for suggestion.

Did the patient experience ischemic-reperfusion injury after restoration of blood flow to the extremities? Did you perform fasciotomies? Was the patient transferred to the ICU after the operation and for how long?

Yes, patient experienced compartment syndrome on the right limb. Fasciotomies were performed after the procedure of blood flow restoration. Yes, of course the patient was transferred in IC for 60 hours.

What was the platelet treatment of the patient after discharge?Viabahn endoprosthesis requires double antiplatelet treatment for at least 3 months. please refer

Done.

Discussion is well written describing the correlation of Covid 19 with thrombosis.

Thank you.

212 line. It is difficult to perform larger studies in patients with AAA treated with EVAR and recently positive Covid 19 patients. Please erase it.

Done.

Thank you again, hoping to have clarified all his interesting doubts.

Best regards,

The Authors.

Reviewer 2 Report

Comments and Suggestions for Authors

Covid-19 infection can lead to many complications, including thromboembolic ones. The presented case is confirmation of such clinical risk, as it concerns thrombosis in an aortic stent-graft. This is a very interesting case, which is presented in an appropriate attractive way. Additionally, the course of the diagnostic and therapeutic procedure is very didactic and carried out correctly. 

Author Response

Dear Reviewer,

thank you very much for your consideration.

For us, Despite the presence of less Sars-Cov2 aggressive forms and increasingly updated vac-cines, continue to investigate still unclear aspects of covid remains fundamental given the still high spread of the virus and the possibility of new pandemics caused by pathogens with characteristics similar to Sars-Cov2.

Best regards,

The Authors.

Reviewer 3 Report

Comments and Suggestions for Authors

This manuscript described a potential case of aortic stent-graft thrombosis in patient with recent COVID-19. Althought the case is interesting, the association between SARS-CoV-2 and thrombosis is difficult to elucidate. 

1. Please add more descripition about SARS-CoV-2 infection in this patient. Why he took exam for SARS-CoV-2 and what the anti-COVID-19 treatment for this patient.

2. Please shorten the description about the clinical course of the management of aortic stent.

3. Please shorten the first five paragraph in the disuccion. Most of them like introduction or literature review. Just discuss  your own findings.

Author Response

Dear Reviewer,

thank you very much for your corrections and advice. The new manuscript has been implemented with all your suggestions.

This manuscript described a potential case of aortic stent-graft thrombosis in patient with recent COVID-19. Althought the case is interesting, the association between SARS-CoV-2 and thrombosis is difficult to elucidate. 

    1. Please add more descripition about SARS-CoV-2 infection in this patient. Why he took exam for SARS-CoV-2 and what the anti-COVID-19 treatment for this patient.

Done, EVAR procedure was performed four days after the resolution of a Sars-Cov2 infection, confirmed by reverse transcriptase-polymerase chain analysis, performed due the presence of a Covid infection cluster in our Surgical Department.

  1. Please shorten the description about the clinical course of the management of aortic stent.

The endovascular surgery part could not be reduced because another reviewer requested additional details.

  1. Please shorten the first five paragraph in the discussion. Most of them like introduction or literature review. Just discuss your own findings.

Thank you.

Thank you again,

Best regards,

The Authors.

Round 2

Reviewer 1 Report

Comments and Suggestions for Authors

Dear Authors 

I would like to thank you for your corrections.Now, you describe in details all the steps you performed with the management of your patient without leaving any gaps in the text. I have not any further comments. Congratulations

Author Response

thank you!

Reviewer 3 Report

Comments and Suggestions for Authors

The authors revise well, so I have no more comment.

Author Response

Thank you!